# The Effects of the Modified Transtheoretical Theory of Stress and Coping (TTSC) Program on Dementia Caregivers’ Knowledge, Burden, and Quality of Life

**DOI:** 10.3390/ijerph182413231

**Published:** 2021-12-15

**Authors:** Worarat Magteppong, Khemika Yamarat

**Affiliations:** College of Public Health Sciences, Chulalongkorn University, Bangkok 10330, Thailand; Worarat.M@student.chula.ac.th

**Keywords:** dementia caregivers, transtheoretical theory of stress and coping, home visit and telephone follow-up, knowledge, burden, quality of life

## Abstract

This quasi-experimental study aimed to examine the effect of the modified transtheoretical theory of stress and coping (TTSC) program on the knowledge, burden, and quality of life of dementia caregivers. The participants comprised 60 caregivers (30 participants in each group) selected via purposive sampling, and the study was conducted between October 2018 and September 2019 in a semi-urban area of central Thailand. The experimental group received an 8-week program, while the comparison group received routine care. A self-administered questionnaire was used to collect data. To analyze the intervention’s effectiveness, repeat measure ANOVA and Mann–Whitney, Friedman, and Dunn’s tests were performed. At the end of the program and again three months after the end of the program, the knowledge and quality of life scores for the experimental group were significantly higher (*p* < 0.05 and *p* < 0.05, respectively) than for the control group. The burden score decreased in the experimental group and increased in the control group in week 8 (*p* < 0.05). There was no statistically significant difference between the groups, as demonstrated by ANOVA (F[1.58] = 2.394; *p* = 0.127). Our findings show that this program had a positive effect on the caregivers’ knowledge and quality of life. However, the program did not affect the caregivers’ burden.

## 1. Introduction

Dementia is a progressive disease, and the available treatments cannot reverse the associated degeneration of brain cells [1]. The incidence of dementia in Thailand has been increasing as the population ages [2]. Elders in the later stages of dementia experience serious problems, such as being disorientated, having difficulty understanding situations, and experiencing an inability to undertake activities and self-care without assistance [3]. More than 80% of elders with dementia need constant care. In addition, elders with dementia need higher levels of care than those without [4]. Overall, 62% of caregivers have reported experiencing a burden associated with caring for people with dementia, and 43% of caregivers have demonstrated a mild to moderate burden [4,5]. Many studies have shown that dementia patient caregivers experience higher levels of stress [6] and a lower quality of life [7] than those caring for physically disabled or elderly patients. A study in one province of northeastern Thailand found that age, occupation, income, relationship, social support, health problems, lifestyle habits, activities of daily life level, cognitive level, sleep quality, and daily hours of providing care were significantly associated with caregiver burden [8]. Additionally, the caregivers of dementia patients with severe behavioral and psychological symptoms showed high levels of caregiver burden [9]. While the caregivers of dementia patients who lacked an understanding of dementia may not have delivered good care and may be at risk of caregiver burden [10], the caregivers who had a higher level of knowledge had better decision making and a lower level of caregiver burden [11]. The caregivers who had a high level of burden had a greater chance of experiencing insomnia [12] and depression [13]. A low education level regarding caregiver burden was a risk factor for reduced quality of life [14], and a high level of caregiver burden caused poor quality of life [15].

As caregiver burden is associated with a high level of stress, the transactional theory of stress and coping (TTSC) was a fundamental theory for the intervention in this study [16]. Equally important were the findings of a previous study in Thailand that the top three caregiver needs were caregiver education and training, telephone lines for caregiver consultations, and special systems in hospitals for dementia patients to have rapid access to doctors [17]. Furthermore, multicomponent interventions that comprise a combination of interventions are reported to be more effective than interventions targeting a single point of caregiver functioning. Likewise, fixed interventions are less beneficial than those tailored to the specific needs of caregivers [6,18]. The effectiveness of multicomponent interventions is reflected in caregiver burden, depression, quality of life, mood, and sense of competence [18], behavioral and psychological symptoms of dementia [19]. The home visit and telephone follow-up program was conducted as an integrated intervention based on the TTSC. After reviewing the studies presented above, this theory was applied to the main intervention activities of the modified TTSC program.

Bangkok (BKK) has the highest number of elderly people in Thailand [20], and many studies have examined interventions to decrease caregiver burden related to dementia in BKK. However, studies of caregiver burden in semi-urban areas of Thailand are rare. Ratchaburi, where our study took place, has one of the top three highest number of dementia patients in the central region of Thailand. Ratchaburi is around 100 kilometers from BKK [21] and is a semi-urban area that lacks education opportunities [22]. Consequently, Ratchaburi’s educational level is lower than that of BKK [23]. Furthermore, the income of dementia caregivers in this province is also lower than in BKK [24]. Therefore, Ratchaburi is an ideal study area for caregiver burden related to dementia in Thailand. In Ratchaburi province, which includes semi-urban and semi-rural communities, the percentage of elderly and dementia patients is higher than the percentage for the whole of Thailand [25].

There were two main hypotheses. Firstly, the post-test and follow up mean scores of caregiver burden in the experimental group are lower than the control group. Secondly, mean scores of knowledges and quality of life would be better in the experimental group compared to the control group at end of program and follow-up. In order to reduce caregiver’s burden and increase quality of life of dementia caregivers, the purpose of this study was to examine the effect of the modified transtheoretical theory of stress and coping (TTSC) program on dementia patient caregivers’ burden, knowledge, and quality of life.

## 2. Materials and Methods

### 2.1. Study Design

This study used a quasi-experimental design in which two groups were assessed prior to the intervention, at the end of the program, and as a follow-up several weeks after the intervention had been completed. These assessments were conducted between October 2018 and September 2019 in a semi-urban area of Ratchaburi province, Thailand.

### 2.2. Study Population and Sample Size

In 2018, the total number of elderly people in Ratchaburi province was 148,011 [20], that is, 16.98% of the total population of the province. This number was higher than the average percentage of elderly people in Thailand. The estimated number of dementia patients was 10,055 [26]. To calculate the sample size, the mean and standard deviation from a previous study [27] were used with a *t*-test for two independent samples (one-tailed test) via the G*power program. The alpha (α) for the test was set at 0.05 to achieve a power of 0.95. After adding a 20% attrition rate, the total sample size was 31 participants per group.

First, Ratchaburi hospital was selected to recruit caregivers since this is the only tertiary hospital with a dementia clinic in this province. Second, 140 medical records were screened for 97 caregivers who live in Ratchaburi province. Third, all 97 caregivers were screened by interview for inclusion criteria, and exclusion criteria. Fourth, dementia clinic service days were randomized to be control day and experimental day. Caregivers were recruited by service day into control and experimental groups until there were 31 in each group. They gave informed consent on the day that they brought a dementia patient or came alone to the dementia clinic from October to December 2018. Inclusion criteria were (1) caregivers who were more than 20 years old, (2) primary caregivers caring for relatives or family members with reported significant memory loss or deterioration in cognitive abilities who were screened by Mini-Mental State Examination-Thai 2002 (MMSE-Thai 2002) and had a diagnosis confirmed by either a neuro-medicine doctor or a psychiatrist for at least 6 months, (3) investing at least 6 hours per day in caregiving activities, (4) had a landline or mobile phone, and (5) planning to remain in the area as the primary caregiver for not less than eight months. Exclusion criteria were (1) a formal caregiver who worked in a nursing home, (2) a patient diagnosed with uncontrolled physical illness or psychiatric illness by a doctor, and (3) a caregivers could not communicate, read, and write in the Thai language.

The total number of caregivers was 62. Due to the fact the clinic was open two-day per month, and for contamination prevention purposes, the caregivers who came on Monday of the 4th week of the month were assigned to the experimental group and those who came on the Monday of the 2nd week of the month were assigned to the control group.

### 2.3. Intervention

The transactional theory of stress and coping (TTSC) was the main theory used to design this intervention, which integrated the results of a previous study in Thailand into three main activities of this multicomponent program. TTSC focuses on the psychological response of a person and deals with how a person copes with stressful situations. The major concepts involved are stress, appraisal, and coping [16].

This intervention aimed to increase caregivers’ knowledge, to reduce caregivers’ burden, and, eventually, to increase caregivers’ quality of life. The intervention consisted of three activities carried out for eight weeks. Caregivers in the control group received routine care.

Three research assistants participated in the study; they are registered nurses and received training from the researchers. The training course included research objectives, the content of a modified TTSC program, knowledge about people with dementia and how to care for them, coping strategies, and counseling techniques. The researcher took the training course three times (3 h/time). The research assistants who passed the training course and evaluated the process of data collection were assigned to be participants for data collection. The researcher observed the research assistants throughout the process of data collection and rechecked the quality of data collection by meeting the research assistants every week.

The process of data collection in each group used the process outlined in Figure 1.

The modified TTSC program was a multicomponent program comprising group education, psychoeducation, and home arrangement through group health education, home visit, and telephone tracking. The program was conducted over 8 weeks as follows (Table 1):

### 2.4. Instruments

The study used three instruments to measure outcomes and the index of item-objective congruence was used by five experts to check the validity of the questionnaires. The questionnaires were tested 30 dementia caregivers who has similar characteristics to the participants in this research. Then, Cronbach coefficient alpha was calculated.

The Dementia Knowledge Assessment, Version 2 [28] was translated into Thai and refined for use in this study with the approval of the tool’s author. The assessment includes 21 questions designed to examine knowledge about dementia via a three-point scale (true, false, or do not know). The maximum knowledge score is 21 points, with correct answers scoring one point and incorrect or unsure answers earning no points. A high score represents a high level of knowledge. The questionnaire showed a Cronbach coefficient alpha of 0.94.

The Thai Burden Interview for Caregivers of Patients with Chronic Illness [29] includes 22 items for the assessment of caregivers’ feelings about the impact of caregiving on their emotional and physical health, functioning, social life, and financial status. Each item is a statement that the caregiver is asked to endorse using a 5-point scale (0 to 4). The total score range is 0–88 points. A high score represents a high burden. The questionnaire showed a Cronbach coefficient alpha of 0.90.

The World Health Organization’s Quality of Life—Thai [30] comprises 26 items that measure four domains (physical health, psychological health, social relationships, and environment) and two items that measure the overall quality of life and general health. It uses a five-point rating scale (1 to 5). The total score range is 26–130 points. A high score represents a high quality of life. The questionnaire showed a Cronbach coefficient alpha of 0.96.

### 2.5. Data Collection

Data were gathered between October 2018 and September 2019. The collection of pre- and post-intervention data took place over 8 weeks on average, and the follow-up took place on average 12 weeks post-intervention.

### 2.6. Data Analysis

Descriptive statistics were used to describe the sociodemographic characteristics, knowledge score, caregiver burden score, and quality of life score and the data were quantified in frequency, percentage, mean, minimum, maximum, and standard deviation. Data analysis was carried out using SPSS version 16. Sociodemographic characteristics between the groups were analyzed in terms of frequencies and percentages. Sociodemographic differences between the two groups were tested using chi-square and Fisher exact tests. Normality was tested by using the Shapiro–Wilk test for knowledge score, caregiver burden, and quality of life at baseline, week 8 and week 20. Knowledge scores and quality of life exhibited non-normal distributions. The caregiver burden was a normal distribution. A repeat measure ANOVA was used to compare the group means for caregiver burden over time. The Mann–Whitney test was used to compare the group means for the dependent variables of knowledge and quality of life across and between the intervention and control groups. Friedman’s and Dunn’s tests were used to compare the group means for the dependent variables of knowledge and quality of life within the groups with significance indicated by a *p*-value < 0.05.

## 3. Results

### Sociodemographic and Characteristic Variables

Sixty-two dementia patient caregivers were recruited into this study, with 31 caregivers each in the experimental and control groups. Two participants were excluded from the study (one participant of each group) because one dementia patient in the experimental group died, and one dementia caregiver in the control group did not follow up and could not be contacted during the follow-up period.

Table 2 presents the demographic characteristics of the experimental and control groups at baseline. The majority of the people in both groups were female, 41–60 years old, married, and worked full-time. The mean ages of the experimental group and the control group were 53.06 and 52.52 years, respectively. The mean incomes of the experimental group and the control group were 11,661.29 and 12,064.52 baht per month, respectively. There were no significant differences between the experimental group and the control group regarding gender, age, marital status, education level, employment status, occupation, relationship to care recipient, length of time as a caregiver, time spent caring, family members that assist in the caring, number of people in the house, or income (*p*-value > 0.05). The majority of the patients’ dementia stage for both groups was moderate. The majority of those in both groups were the daughter (blood relatives) of dementia patients, and they lived with their care recipients.

Table 3 presents the mean and standard deviation of the caregiver burden scores. A comparison of the mean and standard deviation of the caregiver burden scores in the experimental group showed that for the baseline, the mean score was 45.67 ± 12. At the end of the program (week 8), the mean score for caregiver burden decreased to 43.27 ± 10.75 but it increased to 44.00 ± 9.99 in week 20. The baseline score for the control group’s burden was 47.33 ± 13.52; in week 8 the mean score for caregiver burden increased to 49.37 ± 11.84 and increased to 50.07 ± 11.44 in week 20. A comparison of the caregivers’ burden between the groups was conducted before, after, and during a follow-up 3 months after the program’s implementation. The repeated-measure ANOVA results showed no statistically significant difference between the groups (*p* = 0.127).

Table 4 presents the mean and standard deviation of the knowledge and quality of life scores. A comparison of the mean and standard deviation of knowledge scores in the experimental group showed that for the baseline, the mean score was 10.23 ± 1.99. At the end of the program (week 8), the mean score for knowledge increased to 13.63 ± 1.65 and it increased to 13.77 ± 1.63 in week 20. The baseline score for the control group’s knowledge was 10.35 ± 1.89. In week 8, the mean score for knowledge increased from 10.35 to 10.97 ± 2.06 and increased to 11.13 ± 2.11 in week 20. A Friedman test indicated that knowledge was different across time within the experimental group, χ^2^(2) = 55.802, *p* < 0.05 and the control group χ^2^(2) = 17.148, *p* < 0.05. Therefore, there was at least one pair difference.

A comparison of the mean and standard deviation of quality-of-life score in the experimental group showed that for the baseline, the mean score was 85.03 ± 9.80 at the end of the program (week 8); the mean score for quality of life increased to 88.13 ± 8.83 but it decreased to 87.07 ± 8.82 in week 20. The baseline score for the control group’s quality of life was 86.52 ± 9.21. In week 8, the mean score for knowledge decreased from 86.52 to 79.65 ± 7.84 and decreased to 77.87 ± 6.66 in week 20. A Friedman test indicated that quality of life was different across time within the experimental group, χ^2^(2) = 43.130, *p* < 0.05 and control group χ^2^(2) = 46.308, *p* < 0.05. Therefore, there was at least one pair difference.

Table 5 presents a comparison of the knowledge and quality of life scores. For the experimental group, there were significant differences in the knowledge scores between the baseline and week 8 (*p* < 0.05) and the baseline and week 20 (*p* < 0.05). There were significant differences in the quality-of-life scores between the baseline and week 8 (*p* < 0.05) and the baseline and week 20 (*p* < 0.05). However, for the control group, there were significant differences in the knowledge score between the baseline and week 20 (*p* = 0.017) and significant differences in the quality-of-life score between the baseline and week 8 (*p* = 0.001) and the baseline and week 20 (*p* < 0.05).

Table 6 presents a comparison of the knowledge and quality of life scores for the experimental and control groups at baseline, week 8, and week 20. There were no significant differences in the knowledge score between the experimental and control groups at baseline (*p* = 0.748). However, caregivers in the experimental group had a better knowledge score after week 8 (*p* < 0.05) and week 20 (*p* < 0.05) compared to the control group.

For the quality-of-life scores, there were no significant differences between the experimental and control groups at the baseline (*p* = 0.451), but the caregivers in the experimental group had a better quality of life after week 8 (*p* < 0.05) and week 20 (*p* < 0.05) compared to the control group.

## 4. Discussion

This quasi-experimental study aimed to examine the effect of the modified transtheoretical theory of stress and coping (TTSC) program on the knowledge, burden, and quality of life of dementia caregivers. Thirty-one dementia caregivers were selected as the experimental group, and 31 dementia caregivers were selected as the control group. One dementia caregiver in the experimental group dropped out because the dementia patient they cared for passed away, and one dementia caregiver in the control group was lost during the follow up. In total, then, 60 dementia caregivers were entered in the final analysis. The results consisted with research hypothesis that the modified TTSC program significantly increased the scores of knowledges and quality of life in the experimental group, which were higher than in the control group at 8 weeks and 20 weeks. In contrast, the result was not concordant to contrary to research hypothesis. There was no significance difference in the caregiver burden score between the experimental group and the control group. The repeated measure ANOVA result showed no statistically significant difference between the groups.

A comparison of the knowledge scores was undertaken before and after the intervention, as well as at the follow up after the implementation of the modified TTSC program in the experimental group and control group had ended. This study found a significant increase in the knowledge score for the experimental group following the modified TTSC program compared to the control group in week 8 and week 20. This result was consistent with those of other researchers who employed multicomponent programs for dementia caregivers [31,32,33]. The group education activity in this study was similar to boot camp on knowledge [31,32]. Moreover, the multicomponent of this study increase knowledge level as well as the multicomponent psychosocial intervention program [33]. Obviously, the knowledge of the control group continuously increased from baseline to 8 weeks and 20 weeks. This could be explained by the fact that not only group health education but also the telephone follow-up and home visit boosted knowledge required for a specific situation. Additionally, providing a dementia caregiver handbook may increase the knowledge of the caregiver. The knowledge score of the control group had a slow increase from baseline to week 20. This slow increase in the control group’s knowledge score was due to routine care and may have been influenced by other sources such as brochures, media, etc.

In terms of caregiver burden, there was no significant difference in the experimental group’s burden scores following the experimental compared to the control group. This result is consistent with a previous study using a case manager via home visit and telephone calls, where there was no significant difference between the intervention and control group [34]. This study’s results are inconsistent with the results of other researchers who employed a multicomponent program for dementia patient caregivers [33,35]. However, the results found that for the experimental group, the caregivers’ burden scores decreased, while they increased in the control group. In the experimental group, the caregivers’ burden scores decreased after program implementation or at week 8, but the caregivers’ burden score increased at the follow-up and at 20 weeks. Based on the summary of individual stress and coping in the experimental group each week, 26 participants applied problem focus and/or emotional focus to cope but four participants could not cope or solve the problems. Therefore, the multicomponent activities in the first eight weeks of modified TTSC may reduce caregiver burden in an experimental group. Moreover, increased caregiver burden is associated with dementia patients’ increased neuropsychiatric symptoms, behavioral abnormalities, disabilities, and their need for help with daily living [36], and disruptive behaviors [37]. To reduce the caregiver burden, the duration of intervention programs should be extended, and other activities should be added. Planning for the community to continuously take care of dementia caregivers is also important. In addition, other research studies have obtained different results for decreasing caregiver burden using different intervention methods, such as coping strategies, cognitive-behavioral intervention, group discussion, and classroom sessions [35,38].

This study found a significant increase in the quality-of-life scores for the experimental group following the modified TTSC program compared to the control group in week 8 and week 20. The result was consistent with other studies [39,40]. There are four components of quality-of-life measurement (physical health, psychological health, social relationship, and environment). The result of this study pointed out that the psychological health and environment scores changed more than the physical health and social relationship scores. This can be explained by home visit activity that aimed to arrange the home in a way that would maintain the safety of people with dementia outside and inside. Additionally, the modified TTSC program helped the caregivers to define and solve the problems, including coping with their emotions in order to solve the problem. The modified TTSC program may have influenced the quality-of-life components, especially regarding psychological health and environment as explained above

### Strengths, Limitations, and Recommendations

This study has two strengths. First, the research team members who conducted the intervention program work in the research area; therefore, they understand the participants’ context. Second, the study described an outcome at an average of 12 weeks after the end of the program. Therefore, the results showed continuous effects on the outcome even when the participants did not receive the intervention and use the results to plan for implementation. However, there are some limitations. First, the study used a quasi-experimental design; therefore, it could not control all the external co-interventions and confounders. Second, the sample size was small, which may limit the generalizability of the study.

For further studies, it is recommended that (1) activities that reduce caregiver burden should be added to programs and the duration of intervention programs should be extended, (2) repeated studies with larger sample sizes should be undertaken, (3) and the program should be conducted with dementia patient caregivers in other areas in Thailand to confirm its effectiveness and acceptability. For the implementation, (1) health providers should consider including a home visit and a telephone follow-up program practice in protocols for dementia clinics and health-promoting hospitals and (2) the program could be adjusted to be part of routine care for nurses and village health volunteers who take care of dementia caregivers.

## 5. Conclusions

The objective of this study was to determine the effect of the modified TTSC programs on knowledge, caregiver burden, and quality of life among dementia caregivers. Our findings indicate the effectiveness of the modified TTSC programs for dementia patient caregivers to increase their knowledge and quality of life.

## Figures and Tables

**Figure 1 ijerph-18-13231-f001:**
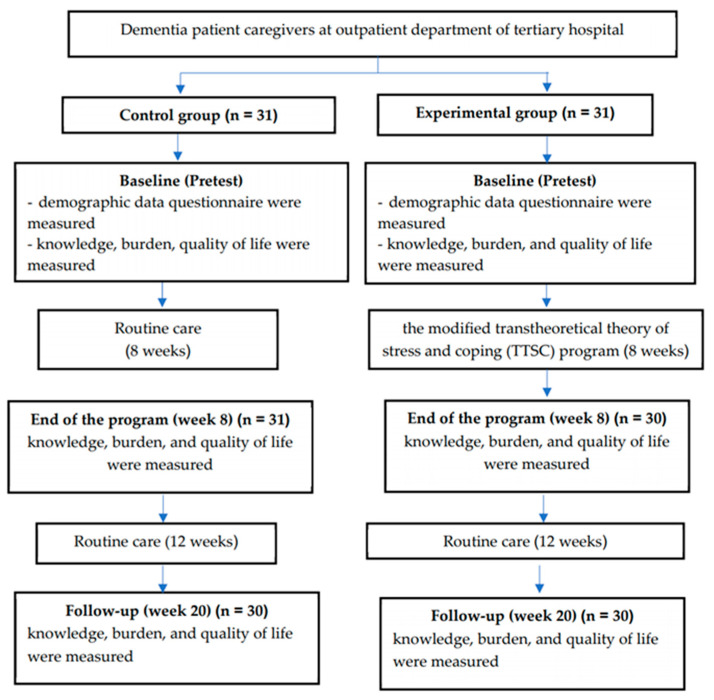
Process of data collection of the study.

**Table 1 ijerph-18-13231-t001:** The activities of the modified transtheoretical theory of stress and coping (TTSC) program.

Week	Objective	Activities and Content	Evaluate
1	To share additional requirements, problems, and methods to solve the problems	Group health education at dementia clinic (30–40 min) (researcher and research assistants per 5–6 participants per group) Step 1: Introduction and building confidence: The researcher introduces the details of the program and builds caregivers’ confidence that their problems will be heard and handled. Caregivers cannot avoid the problems; however, we can handle the problems. Step 2: Caregivers share about the caring experience. - “How can you take good care of people with dementia?” - “What are the challenges of caring for people with dementia?” Step 3: Caregivers’ summary topic - “What patient care do you do well and what challenges do you have with patient care?” - “What is gained from participating in group health education?” - “What help do you need from the researcher?” Step 4: An after group-education caregiver dementia handbook was provided to the experimental group.	Sharing additional requirements, caregiver’s problems, and methods to solve the problems
2	To explore self-assessment and home arrangement	Home visit (45–90 min) Major concept: Stress The researcher encourages the caregivers to explore a self-assessment of their stressors. The caregivers’ stressors are the challenges of caring for people with dementia. The researcher helps the caregivers explore a self-assessment of the problems of caring for people with dementia that significantly contribute to their response to stress. The researcher assesses the safety of the home environment. Major concept: Appraisal The researcher assesses the dementia caregivers. “What does this stressor and/or situation mean?” The researcher assists caregivers in classifying the stressor or situation. - “How can it influence you?” - “Are the problems a threat, a challenge or a harm/loss?” The researcher asks, “Do you have resources available for caring for people with dementia?” The researcher supports a positive appraisal by encouraging the caregivers to feel as though, “I can take care of people with dementia by myself”, “I will try to care for people with dementia whether my chances of success are high or not”, and “If I can’t care for people with dementia, I can always try another method”. Major concept: Coping The researcher helps the caregivers overcome stress by using coping strategies. - Problem-focused: The researcher helps the caregivers define the problems and provides education tailored to the specific needs of the caregivers for solving problems of caring for people with dementia. Example: home arrangement for maintaining safety outside and inside for people with dementia, communication techniques, guidelines for preventing accidents. - Emotion-focused: The researcher teaches and counsels the caregivers how to control their emotional response to the problem. Example: exercise, muscle relaxation, deep breathing.	- Explore self-assessment - Ensure safety of home environment arrangement - Offer a positive cognitive appraisal - Method used to solve problems
3–7	To re-explore self-assessment and support a positive appraisalTo help the caregiver overcome stress by using coping strategies	Telephone follow-up (once a week) (15–30 min) The caregivers receive telephone contact once a week. This telephone tracking focuses on providing education that is tailored to the specific needs of the caregiver. Moreover, the intervention guides the dementia caregivers to use specific coping strategies based on the transtheoretical model. Major concept: Stress The researcher encourages the caregivers to re-explore the self-assessment of their stimuli stressors that significantly contribute to their response to stress. The researcher assesses a dementia caregiver’s status and identifies positive and negative changes since the last contact. Major concept: Appraisal The researcher assesses the dementia caregivers’ problem. “How does the stressor influence you?” The researcher assesses and takes notes of any changes in each key area of functioning such as health functioning, mood, family support. The researcher reinforces the need to appraise and reappraise these issues. The researcher helps caregivers classify the stressor or situation. - “How can it influence you?” - “Are the problems a threat, a challenge or a harm/loss?” The researcher asks, “Do you have resources available for caring for people with dementia?” The researcher supports a positive appraisal by encouraging the caregiver to feel as though, “I can take care of people with dementia by myself”, “I will try to care for people with dementia whether my chances of success are high or not”, and “If I can’t care for people with dementia, I can always try another method”. Major concept: Coping The researcher helps the caregivers overcome stress by using coping strategies. - Problem-focused: The researcher helps the caregivers define the problems and provides education tailored to the specific needs of the caregivers for solving problems of caring for people with dementia. Example: home arrangement for maintaining safety outside and inside for people with dementia, communication techniques, guidelines for preventing accidents. - Emotion-focused: The researcher teaches and counsels the caregivers how to control their emotional response to the problem. Example: exercise, muscle relaxation, deep breathing.	- Explore self-assessment - A positive cognitive appraisal - Method used to solve problems
8	To re-explore self-assessment and support positive appraisal To help the caregivers overcome stress by continuing to develop and utilize adaptive coping strategies To inform caregivers regarding termination of the intervention	Home visit (45–90 min) Major concept: Stress The researcher encourages the caregivers to re-explore self-assessment and summarize their stimuli stressors that significantly contribute to their response to stress. The researcher re-evaluates the safety of the home environment and summary about home arrangement. Major concept: Appraisal - The researcher supports a positive appraisal by encouraging caregivers to feel as though, “I can take care of people with dementia by myself”, “I can provide good care for people with dementia”, “I can solve the problems of caring people with dementia”. Major concept: Coping - The researcher asks caregivers to describe how they handled difficulties over the last month. - The researcher encourages the caregivers to continue to develop and utilize adaptive coping strategies. - The researcher recommends a support service or health care team (dementia clinic at Ratchaburi hospital) to continue the care for people with dementia and dementia caregivers.	- Explore self-assessment - A positive cognitive appraisal - Problem-solving ability - Caregivers’ knowledge - Caregivers’ burden - Caregivers’ quality of life
9–20	Reappraisal	none	- Caregivers’ knowledge - Caregivers’ burden - Caregivers’ quality of life

The control group were given routine care that was provided by the dementia clinic nurses. Caregivers explored the problems of dementia patients and also were advised on how to address the problems by the nurses in the clinic. Moreover, the control group received dementia handbook at 20 weeks.

**Table 2 ijerph-18-13231-t002:** Baseline demographic characteristics of the experimental and control groups.

Characteristics	Experimental Group (*n* = 31)		Control Group (*n* = 31)		*p*-Value
	*n*	%	*n*	%	
Gender					
Male	8	25.8	4	12.9	
Female	23	74.2	27	87.1	0.19 ^a^
Age					
20–40 years	2	6.5	3	9.7	
41–60 years	24	77.4	24	77.4	
≥60 years	5	16.1	4	12.9	0.86 ^b^
Marital status					
Single	11	35.5	7	22.6	
Married	17	54.8	17	54.8	
Divorced/Separated/Widowed	3	9.7	7	22.6	0.41 ^a^
Relationship to care recipient					
Spouse	5	16.1	5	16.1	
Son (blood relatives)	6	19.4	3	9.7	
Daughter (blood relatives)	15	48.4	18	58.1	
Relative	3	9.7	2	6.5	
Friend/Adopted child/Neighbor	2	6.5	3	9.7	0.87 ^b^
Education level					
Primary school	12	38.7	8	25.8	
High school	11	35.5	15	48.4	
Bachelor’s degree	8	25.8	8	25.8	0.49 ^a^
Employment Status					
Full time	17	54.8	17	54.8	
Part time	3	9.7	4	12.9	
Not Employed/Retired	11	35.5	10	32.3	0.91 ^b^
Length of time as a caregiver					
0–3 years	13	41.9	6	19.4	
3.1–6 years	10	32.3	10	32.3	
6.1–9 years	5	16.1	11	35.5	
More than 9 years	3	9.7	4	12.9	0.17 ^b^
Time for caring (hour/day)					
6–12 h	29	93.6	31	100	
More than 12 h	2	6.4	0	0	0.26 ^b^
Clinical Dementia Rating (CDR) ^c^					
Mild dementia	8	25.8	8	25.8	
Moderate dementia	18	58.1	17	54.8	
Severe dementia	5	16.1	6	19.4	0.94 ^a^

Notes: ^a^ Chi-square test, ^b^ Fisher exact test, ^c^ There is a five-point rating scale of CDR. 0 = no impairment, 0.5 = questionable dementia, 1 = mild dementia, 2 = moderate dementia, and 3 = severe dementia.

**Table 3 ijerph-18-13231-t003:** Mean and standard deviation of caregiver burden score. (*n* = 30).

Outcome	Time	Experimental Group		Control Group	
		Mean	SD	Mean	SD
Caregiver burden	Baseline	45.67	12.0	47.33	13.52
	Week 8	43.27	10.75	49.37	11.84
	Week 20	44.00	9.99	50.07	11.44

**Table 4 ijerph-18-13231-t004:** Mean and standard deviation of knowledge and quality of life and comparison of knowledge and quality of life scores across time (*n* = 30).

Outcome	Baseline		Week 8		Week 20		Test Stat	df	*p*
	Mean	SD	Mean	SD	Mean	SD			
Knowledge									
Experimental group	10.23	1.99	13.63	1.65	13.77	1.63	55.802	2	<0.05 *
Control group	10.35	1.89	10.97	2.06	11.13	2.11	17.148	2	<0.05 *
Quality of life									
Experimental group	85.03	9.80	88.13	8.83	87.07	8.82	43.130	2	<0.05 *
Control group	86.52	9.21	79.65	7.84	77.87	6.66	46.308	2	<0.05 *

* *p* < 0.05.

**Table 5 ijerph-18-13231-t005:** Comparison of knowledge and quality of life scores for the baseline, at the end of the program (week 8), and at the follow-up (week 20) (*n* = 30).

Outcome	Group	Time (Within Group)	Test Stat	Sd. Error	Sd. Test Stat	*p*-Value
Knowledge	Intervention	Baseline–Week 8	−1.383	0.258	−5.358	<0.05 *
		Baseline–Week 20	−1.517	0.258	−5.874	<0.05 *
		Week 8–Week 20	−0.133	0.258	−0.516	1.000
	Control	Baseline–Week 8	−0.583	0.258	−2.259	0.072
		Baseline–Week 20	−0.717	0.258	−2.776	0.017 *
		Week 8–Week 20	−0.133	0.258	−0.516	1.000
Quality of life	Intervention	Baseline–Week 8	−1.467	0.258	−5.680	<0.05 *
		Baseline–Week 20	−0.933	0.258	−3.615	<0.05 *
		Week 8–Week 20	0.533	0.258	2.066	0.117
	Control	Baseline–Week 8	−0.933	0.258	3.615	0.001 *
		Baseline–Week 20	1.517	0.258	5.874	<0.05 *
		Week 8–Week 20	0.583	0.258	2.259	0.072

* *p* < 0.05.

**Table 6 ijerph-18-13231-t006:** Comparison of knowledge and quality of life scores between the experimental and control groups for the baseline, at the end of the program (week 8), and at the follow-up (week 20) (*n* = 30).

Outcome	Time	Experimental Group		Control Group		Mann-Whitney U	df	*p*-Value
		Mean	SD	Mean	SD			
Knowledge	Baseline	10.23	1.99	10.35	1.89	458.00	1	0.748
	Week 8	13.63	1.65	10.97	2.06	140.00	1	<0.05 *
	Week 20	13.77	1.63	11.13	2.11	146.50	1	<0.05 *
Quality of life	Baseline	85.03	9.80	86.52	9.21	427.00	1	0.451
	Week 8	88.13	8.83	79.65	7.84	218.50	1	<0.05 *
	Week 20	87.07	8.82	77.87	6.66	171.50	1	<0.05 *

* *p* < 0.05.

## Data Availability

Data supporting the finding in this study are included within the article.

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
