# Peer review of "The Effects of the Modified Transtheoretical Theory of Stress and Coping (TTSC) Program on Dementia Caregivers’ Knowledge, Burden, and Quality of Life"

_ijerph, 2021, doi:10.3390/ijerph182413231_

Round 1

Reviewer 1 Report

It is a great pleasure for me to have the opportunity to review your paper.

It would be valuable to focus on burden of dementia caregivers.

I have several comments as follows.

1) In the title and purpose of the study, it is stated that the effect of home visits and telephone follow-up will be investigated. However, group health education (week 1) was provided only to the experimental group. Therefore, it seems that what this study is evaluating is the effect of group education + home visits + telephone follow-up.

If the authors were to conduct the study under the current title and purpose of the study, they should have designed the study to include group education for both groups, home visits and telephone follow-up for the intervention group, and routine care for the control group.

In view of the above, I recommend that the title and purpose of the study be revised.

2) Regarding the inclusion criteria of "significant memory loss or deterioration in cognitive abilities" (lines 101-102), I suggest that you need to add information about who diagnosed it and by what measure.

3) Details of routine care should also be described.

4) For replicability, it is desirable to add the number of participants per group education session, the number of facilitators (staff) in relation to the number of participants, and the duration of the session.

 It would be desirable to add information on per telephone or average time.

5) Regarding Table 1, I suggest adding an explanation of the classification criteria for Stage of dementia.

6) Line 201 mentions daughter. In relation to this, in Table 1, are all Females daughters (blood relatives)? If there are previous studies that suggest that there is a difference in the sense of burden depending on whether the patient is a blood relative or a daughter or son in law, is it not also necessary to state whether the patient is a blood relative or not?

7) Regarding lines205-210 of the results, it would be better to demonstrate them not only in text but also in graphs (or at least in tables).

8) Line 209, "After implementing the program", might give a misleading impression that the intervention was also carried out in the control group. The phrase should be changed.

9) Table 2 is difficult to understand. It would be needed to add some explanation for readers.

Author Response

Response to reviewers

Manuscript ID: ijerph-1451747

Paper title: “THE EFFECTS OF THE MODIFIED TRANSTHEORETICAL THEORY OF STRESS AND COPING (TTSC) PROGRAM ON DEMENTIA CAREGIVERS’ KNOWLEDGE, BURDEN, AND QUALITY OF LIFE.”

Dear reviewers,

Thank you very much for giving us the opportunity to revise this manuscript. We have carefully read and revised the manuscript accordingly. Below is the response to the comments. Please refer to our revision. Please kindly let us know if there is other issue need to be addressed.

Comment

Response

1) In the title and purpose of the study, it is stated that the effect of home visits and telephone follow-up will be investigated. However, group health education (week 1) was provided only to the experimental group. Therefore, it seems that what this study is evaluating is the effect of group education + home visits + telephone follow-up.

If the authors were to conduct the study under the current title and purpose of the study, they should have designed the study to include group education for both groups, home visits and telephone follow-up for the intervention group, and routine care for the control group.

In view of the above, I recommend that the title and purpose of the study be revised.

1) Title was revised as following “THE EFFECTS OF THE MODIFIED TRANSTHEORETICAL THEORY OF STRESS AND COPING (TTSC) PROGRAM ON DEMENTIA CAREGIVERS’ KNOWLEDGE, BURDEN, AND QUALITY OF LIFE.”

Thank you very much for your suggestion which made me review the title again. However, it could be the long title if I add every activities of intervention into the title. It would be better to shorten the title as above but still be cover all of your suggestion. Because TTSC was modified by researcher to create activities in intervention as following group education, home visit, and telephone follow.

The purpose of this study is revised as following in abstract, introduction and conclusion parts as following

“examine the effect of the modified transtheoretical theory of stress and coping (TTSC)”.

2) Regarding the inclusion criteria of "significant memory loss or deterioration in cognitive abilities" (lines 101-102), I suggest that you need to add information about who diagnosed it and by what measure.

2) Revised page 3 line 105-107 as following

“who were screened by Mini-Mental State Examination-Thai 2002 (MMSE-Thai 2002) and had a diagnosis confirmed by either a neuro-medicine doctor or the psychiatrist”

3) Details of routine care should also be described.

3) Revised page 7 line 150-151 as following “The control group were given routine care that was provided by the dementia clinic nurses. Caregivers explored the problems of dementia patients and also were advised on how to address the problems by the nurses in the clinic.”.

4) For replicability, it is desirable to add the number of participants per group education session, the number of facilitators (staff) in relation to the number of participants, and the duration of the session.

 It would be desirable to add information on per telephone or average time.

4) Added the details of suggestion in the table 1 page 4-7.

5) Regarding Table 1, I suggest adding an explanation of the classification criteria for Stage of dementia.

5) Revised Table 2 (Table 1) page 9

Stage of dementia was change into Clinical Dementia Rating (CDR) and the explanation is the bottom line of this table as following “there is a five-point rating scale of CDR. 0 = no impairment, 0.5 = questionable dementia, 1= mild dementia, 2 = moderate dementia, and 3 = severe dementia.”

6) Line 201 mentions daughter. In relation to this, in Table 1, are all Females daughters (blood relatives)? If there are previous studies that suggest that there is a difference in the sense of burden depending on whether the patient is a blood relative or a daughter or son in law, is it not also necessary to state whether the patient is a blood relative or not?

Added this variable into table 2 to make it clearer.

7) Regarding lines205-210 of the results, it would be better to demonstrate them not only in text but also in graphs (or at least in tables).

Added the table 2 page 10

8) Line 209, "After implementing the program", might give a misleading impression that the intervention was also carried out in the control group. The phrase should be changed.

Revised page 10 line 238-240 “The baseline score for the control group’s burden was 47.33 ± 13.52; in week eight the mean score for caregiver burden increased to 49.37 ± 11.84 and increased to 50.07 ± 11.44 in week 20.”

9) Table 2 is difficult to understand. It would be needed to add some explanation for readers.

Revised page 10 “A comparison of the caregivers’ burden between the groups was conducted before, af-ter, and during a follow-up three months after the program’s implementation. The repeated- measure ANOVA results showed no statistically significant difference between the groups (p = 0.127).”

Best regards

Mrs.Worarat Magteppong

Assoc. Prof. Dr. Khemika Yamarat

Reviewer 2 Report

Thank you for the opportunity to review this paper. While the topic is of interest in its current form it will require more work before publication. There are a number of areas that require rewriting or clarification. I will comment on these areas section by section.

Introduction.

The introduction is easy to read, however did not extend existing knowledge on this topic. It should include more update references.

After the purpose statement, please provide a hypothesis for what the authors think the results will yield.

Methods
Some important information appears to be presently omitted from the methods section. Further description of the sampling procedure would be helpful for the reader. The analysis process is a bit unclear.

Some important information also appears to be presently omitted from the methods and results section. Have you tested the reliability of your data? If yes, please include the results.

Discussion
In general, the first paragraph of the discussion should at least state which hypotheses were supported. Then the authors should follow with how their results compare with similar data, and what the authors results adds to the literature (different / unique aspects of the data). Several points are made in the discussion, but it is not clear to this reviewer how results from the current study are novel or add to the literature.

The authors shortly discuss several possible explanations for the findings.

Author Response

Response to reviewers

Manuscript ID: ijerph-1451747

Paper title: “THE EFFECTS OF THE MODIFIED TRANSTHEORETICAL THEORY OF STRESS AND COPING (TTSC) PROGRAM ON DEMENTIA CAREGIVERS’ KNOWLEDGE, BURDEN, AND QUALITY OF LIFE.”

Dear reviewers,

Thank you very much for giving us the opportunity to revise this manuscript. We have carefully read and revised the manuscript accordingly. Below is the response to the comments. Please refer to our revision. Please kindly let us know if there is other issue need to be addressed.

Comment

Response

Introduction.

The introduction is easy to read, however did not extend existing knowledge on this topic. It should include more update references.

After the purpose statement, please provide a hypothesis for what the authors think the results will yield.

Introduction.

Explanation 1 I have reviewed the updated studies and added these into the references (6,19). I removed the outdated references from these manuscript.

Explanation 2 I revised the title and purpose of this study by using TTSC as the principle of intervention.

Explanation 3 I think research hypotheses are very importance; however, it will take a longer manuscript. Therefore, I decided not to show research hypotheses in the introduction. Here are research hypotheses of full thesis. 1) The post-test and follow up scores of knowledge and quality of life are higher than baseline in the experimental group.  2) The post-test and follow up scores of caregiver burden is lower than the baseline in the experimental group. 3) The post-test and follow up scores of knowledge and quality of life in the experimental group are higher than in the control group. 4) The post-test and follow up scores of caregiver burden in the experimental group are lower than the control group.

Methods
Some important information appears to be presently omitted from the methods section. Further description of the sampling procedure would be helpful for the reader. The analysis process is a bit unclear.

Method

Added sampling in page 2-3 already.

Revised analysis process as following page 8. “Descriptive statistics were used to describe the sociodemographic characteristics, knowledge score, caregiver burden score, and quality of life score and the data were quantified in frequency, percentage, mean, minimum, maximum and standard devia-tion. Data analysis was carried out using SPSS version 16. Sociodemographic charac-teristics between the groups were analyzed in terms of frequencies and percentages. Sociodemographic differences between the two groups were tested using chi-square and Fisher exact tests. Normality was tested by using the Shapiro-Wilk test for knowledge score, caregiver burden, and quality of life at baseline, week 8 and week 20. Knowledge scores and quality of life exhibited non-normal distributions. The caregiver burden was a normal distribution. A repeat measure ANOVA was used to compare the group means for caregiver burden over time. The Mann–Whitney test was used to compare the group means for the dependent variables of knowledge and quality of life across and between the intervention and control groups. Friedman’s and Dunn’s tests were used to compare the group means for the dependent variables of knowledge and quality of life within the groups with significance indicated by a p-value < 0.05.”

Some important information also appears to be presently omitted from the methods and results section. Have you tested the reliability of your data? If yes, please include the results.

Added the details of suggestion page 8 line 157-159 “The questionnaires were tested 30 dementia caregivers who has similar characteristics to the participants in this research. Then, Cronbach coefficient alpha was calculated.”

Discussion
In general, the first paragraph of the discussion should at least state which hypotheses were supported. Then the authors should follow with how their results compare with similar data, and what the authors results adds to the literature (different / unique aspects of the data). Several points are made in the discussion, but it is not clear to this reviewer how results from the current study are novel or add to the literature.

The authors shortly discuss several possible explanations for the findings.

Discussion

Revised Page12-14

Discussion part was extensive revised in order to clearer correct and concise.

Best regards

Mrs.Worarat Magteppong

Assoc. Prof. Dr. Khemika Yamarat

Reviewer 3 Report

This paper shows that after eight weeks of education for care takers with dementia, their knowledge of dementia improved and their quality of life increased significantly.
The results are natural and seem correct, but as a scientific study the paper seems to have little value.
As they themselves note (lines 321-324), for one thing, the number of subjects was too small to detect details of educational effects (which parts were more effective, etc.), so they could only detect an overall increase in knowledge, and the same is true for quality of life.
The second problem is that the assignment of the control and intervention groups was not randomly chosen but using two different groups existed before the start. As the authors note, this makes it impossible to rule out the possibility that the two populations may have different backgrounds and unintended exposures to external factors. In particular, the significant changes in quality of life in the control group during the observing period are quite strange but remained unexplained.
The authors seem to infer that increased knowledge of dementia is responsible for the increase in quality of life, but to see this, it is necessary to provide at least the same supportive program to the control group as to the intervention group, without education about dementia.
though it is understandable that this is a study of real people with unavoidable  limitations, the conclusions that can be drawn are not very new and weak as an evidence.

The tables seem to be poorly explained, as if they were the output of statistical software. It should be clear which are the raw scores and variances and which are the test results.

Author Response

Response to reviewers

Manuscript ID: ijerph-1451747

Paper title: “THE EFFECTS OF THE MODIFIED TRANSTHEORETICAL THEORY OF STRESS AND COPING (TTSC) PROGRAM ON DEMENTIA CAREGIVERS’ KNOWLEDGE, BURDEN, AND QUALITY OF LIFE.”

Dear reviewers,

Thank you very much for giving us the opportunity to revise this manuscript. We have carefully read and revised the manuscript accordingly. Below is the response to the comments. Please refer to our revision. Please kindly let us know if there is other issue need to be addressed.

Comment

Response

This paper shows that after eight weeks of education for care takers with dementia, their knowledge of dementia improved and their quality of life increased significantly.
The results are natural and seem correct, but as a scientific study the paper seems to have little value.
As they themselves note (lines 321-324), for one thing, the number of subjects was too small to detect details of educational effects (which parts were more effective, etc.), so they could only detect an overall increase in knowledge, and the same is true for quality of life.

Explanation 1 Thank you for your opinion. This made me and my colleagues reviewed sample size and sampling again. This part was revised in the page 2-3 line 92-95

At first, the power use value 0.80 and the sample size in each group was 15. However, the intervention of previous study was only telephone call. This study add up with group education and home visit therefore 0.95 was used as the power in G*Power. Finally, the sample size in each group was 26 adding 20% attrition. Total number of participants in each group was 31. Please educate me if you mention that it is small number.

The second problem is that the assignment of the control and intervention groups was not randomly chosen but using two different groups existed before the start. As the authors note, this makes it impossible to rule out the possibility that the two populations may have different backgrounds and unintended exposures to external factors.

Explanation 2 Although the control and experimental groups were not individually randomized, the participants in the control and experimental group were not significant difference in term of demographic characteristics.

In particular, the significant changes in quality of life in the control group during the observing period are quite strange but remained unexplained.

The authors seem to infer that increased knowledge of dementia is responsible for the increase in quality of life, but to see this, it is necessary to provide at least the same supportive program to the control group as to the intervention group, without education about dementia.
though it is understandable that this is a study of real people with unavoidable  limitations, the conclusions that can be drawn are not very new and weak as an evidence.

Revised Page 13-14

Discussion part especially quality of life had major improvement. Please file attachment(annex) if you want the caregivers’ data, but I don’t want to publish it.

Each component of quality-of-life measurement showed different change. The explanation is in discussion part.

The tables seem to be poorly explained, as if they were the output of statistical software. It should be clear which are the raw scores and variances and which are the test results.

Explanation 3 The tables and description were revised.

Best regards

Mrs.Worarat Magteppong

Assoc. Prof. Dr. Khemika Yamarat

Round 2

Reviewer 3 Report

I appreciate the authors effort for revision and, while I don't think this study is perfect yet, I think it is worth publishing as informative and I accept it, subject to the agreement of other reviewers and editor.

Author Response

Thank you for comments!